# Multi-Channel Neural Recording Implants: A Review

**DOI:** 10.3390/s20030904

**Published:** 2020-02-07

**Authors:** Fereidoon Hashemi Noshahr, Morteza Nabavi, Mohamad Sawan

**Affiliations:** 1Polystim Neurotech. Lab., Department of Electrical Engineering, Polytechnique Montreal, Montreal, QC H3T 1J4, Canada; morteza.nabavi@polymtl.ca (M.N.); mohamad.sawan@polymtl.ca (M.S.); 2School of Engineering, Westlake University, Hangzhou 310024, China; 3Institute of Advanced Study, Westlake Institute for Advanced Study, Hangzhou 310024, China

**Keywords:** neural recording implant, brain–machine interface, analog front-end, neural amplifier, low-noise amplifier, chopper stabilization technique, compressive sensing

## Abstract

The recently growing progress in neuroscience research and relevant achievements, as well as advancements in the fabrication process, have increased the demand for neural interfacing systems. Brain–machine interfaces (BMIs) have been revealed to be a promising method for the diagnosis and treatment of neurological disorders and the restoration of sensory and motor function. Neural recording implants, as a part of BMI, are capable of capturing brain signals, and amplifying, digitizing, and transferring them outside of the body with a transmitter. The main challenges of designing such implants are minimizing power consumption and the silicon area. In this paper, multi-channel neural recording implants are surveyed. After presenting various neural-signal features, we investigate main available neural recording circuit and system architectures. The fundamental blocks of available architectures, such as neural amplifiers, analog to digital converters (ADCs) and compression blocks, are explored. We cover the various topologies of neural amplifiers, provide a comparison, and probe their design challenges. To achieve a relatively high SNR at the output of the neural amplifier, noise reduction techniques are discussed. Also, to transfer neural signals outside of the body, they are digitized using data converters, then in most cases, the data compression is applied to mitigate power consumption. We present the various dedicated ADC structures, as well as an overview of main data compression methods.

## 1. Introduction

In the past decade, researchers have worked on the brain to understand its functions and monitor the brain’s electrical signals to research, diagnose and treat its disorders, as well as to utilize these signals to control artificial limbs. Brain–machine interfaces (BMIs) can serve people with different clinical disorders. For example, researchers have implemented robotic limbs [1,2], speech synthesizers [3], and human neuroprosthetic control of computer cursors [4,5,6], utilizing less than 300 electrodes [7]. In addition, monitoring the bio-potential signals is a fundamental and vital part of a medical diagnostics system. For this purpose, patients are generally connected to a massive bio-potential acquisition equipment. However, this limits the patient’s daily routine on the one hand, and on the other hand requires the long-term arduous monitoring of diagnostics [8]. One of the most promising solutions is to use neural recording implants as a part of BMI systems, which are in high demand and are being developed and improved as technology develops.

Inability to record from large numbers of neurons has limited the development of BMI. Noninvasive methods are capable of recording millions of neurons through the skull, however this signal is nonspecific and distorted [9,10]. Utilizing electrodes placed on the surface of the cortex, an invasive method, records proper signals. However, the disadvantage of this is not being able to record deep in the brain and they average the activity of thousands of neurons [11]. Invasive techniques have been utilized by some BMIs. This is because recording single action potentials from neurons in distributed, functionally-linked ensembles are necessary for the most accurate readout of neural activities [7]. Therefore, increasing the spatial resolution and the number of electrodes are essential for developing BMI.

The implementation of a neural recording implant is multi-disciplinary, as it involves the various scientific fields such as electronics, medical, materials, electrodes, and system integration. Increasing the number of electrodes and, consequently, the number of channels (in the range of thousands), creates new challenges for neural recording in the various fields mentioned. Microelectrode technology is not appropriate for these large-scale recordings [12]. Recently, Neuralink Company has built arrays of small and flexible electrodes (3072 electrodes per array), which have enabled thousands of channel recordings [13]. In the microelectronics field, large-scale recordings create many challenges with regards to decreasing the power consumption and chip area.

In the design of the neural recording implants, the two constraints, the power consumption and chip area, should be addressed. Implantable circuits should consume very low power to avoid any damage to the surrounding tissue due to generated heat. Additional challenges in the design of the analog front-end (AFE) of the neural recording systems arise in advanced and scaled technologies. The main reason is due to the short-channel effects of MOS transistors. These effects in the MOS down-scaled technologies decrease the transconductance ($g_{m}$) of the transistor on one hand and on the other hand increase the gate leakage current, the flicker, and thermal noise power of an MOS transistor. This creates challenges in the design of the high gain and low noise neural amplifier, which will be explained in this paper.

Although there are few reviews on neural recording microsystems and low-frequency low-noise amplifiers [14,15,16], a comprehensive review focused on large-scaled neural recording implants and their challenges was not found in any prior-art publication. To help readers to get involved with the state-of-the-art and challenges of this rapidly growing field, led us to write this paper.

The remainder of this paper is organized as follows. Section 2 reviews the different types of neural signals and their properties. Section 3 presents the essential neural recording architectures in the literature. Section 4 surveys the neural amplifiers. The neural amplifiers are the most challenging part of a neural recording implants. They must be compact, high gain, low power, and low noise amplifiers. To satisfy these constraints in the design of the neural amplifiers, various topologies and techniques are proposed which are presented in the subsections of Section 4. Section 5 covers the ADCs that are suitable for neural implant applications based on their various architectures. Finally, Section 6 discusses the data compression methods in neural recording systems and Section 7 presents the conclusion.

## 2. Neural Signals

The electrical activities of the brain can be recorded through three different methods: (1) from the scalp: its corresponding signal is electroencephalogram (EEG); (2) the surface of the brain, which extracts an electrocorticography (ECoG) or intracranial electroencephalography (iEEG) signal; and (3) within the brain, which captures the extracellular activities of neurons. The extracted signals from these methods have a frequency range of a few mHz to 10 kHz and their amplitude is at the range of 20 μV to 10 mV [17].

In the first method, surface electrodes can be used to un-invasively measure the biopotentials of EEG on the scalp. In contrast, in the second method, the electrodes can be placed directly on the brain surface invasively to record electrical activity from the cerebral cortex. The brain signals provided with this method have a dramatically high signal-to-noise ratio (SNR) and are less sensitive to artifacts than EEG. Furthermore, these signals have high spatial and temporal resolution. Capturing the signal inside the body by utilizing implantable electrodes is the most effective approach for direct control of prosthetic devices [18].

In the third method, a sharp biocompatible microelectrode is utilized to perform bioelectrical recordings invasively inside the body. By penetrating microelectrodes into the brain, the bioelectrical activity that is transmitted along the axon of a neuron can either be measured intracellularly in a single neuron or extracellularly from the brain [19]. The extracellular action potentials (APs) generated by depolarization of the membrane of the neuron are at the range of 100 Hz to 10 kHz and their duration is a few milliseconds. The number of occurrences of the APs are between 10 to 120 times per second. The distance between the active neuron and the recording electrode determines the amplitude of the extracellular APs, which are between 50 μVpp to 500 μVpp [20]. In the following, the extraction procedure of the APs (spikes) and the local field potential (LFP) is explained. LFP is the mean field potential generated by neurons in the vicinity of the electrode.

The neural activities are first amplified after sensing, then are low-pass filtered to obtain the LFP and are also high-pass filtered to identify the activity of single neurons by using spike detection and performing sorting algorithms [21]. The LFP includes lower-frequency neural waveforms in the range of mHz to 200 Hz with an amplitude of 500 μVpp to 5 mVpp. These potentials are used for brain interfacing applications and carry complementary important information [22,23]. High-pass filtering extracts the spikes of the nearby neurons on top of background activity. Amplitude threshold methods are used to detect these spikes. In the next step, the features of the spikes are extracted and sorted accordingly. Note that the SNR of the neurons located at a distance of 50–100 μm from the electrode is large enough to easily distinguish the activity of each single unit [24,25].

For neurons located between 100 μm to approximately 150 μm far from the electrode, spikes can still be detected but their shapes are masked by the noise. These signals are grouped in a ‘multi-unit’ cluster. However, neurons farther than 150 μm from the tip of the electrode cannot be detected and are added to the background noise.

In addition to the extracellular method, the intracellular procedure is explained in [26]. In this method, a sharp glass micropipette penetrates into a neuron of a slice of a brain in the laboratory for neuroscience research. The neurons of the brain slice die after few hours of recording. A metal electrode located inside the micropipette and in contact with electrolyte is connected to an amplifier. The amplitude of intracellular APs is in the range of 10 to 70 mVpp. The neural recording parameters for different signal modalities are summarized in Table 1.

## 3. Neural Recording Architectures

The multi-channel neural recording implants are designed in three main architectures in the literature and are shown in Figure 1. The most common architecture illustrated in Figure 1a exploits one analog to digital converter (ADC) that is shared among all channels. Each channel has a neural amplifier, and the neural signal of each channel is passed to the ADC through an analog multiplexer. The performance of this multiplexing method, which is also referred to as time division multiplexing (TDM) method in analog domain, becomes limited when the number of channels increase dramatically. In order to improve the neural recording spatial resolution, the number of channels increase. This results in a higher sampling frequency of the ADC and the multiplexer which in turn increases the power consumption of the ADC and the driving buffers. Since analog signals are more prone to distortion due to cross-talk noise in analog multiplexers compared to digital signals, careful design considerations have to be accounted for the design process. An example of this design is presented in [27].

The architecture shown in Figure 1b utilizes one ADC for each channel. Due to the low bandwidth of neural signals, the sampling frequency and the power consumption of the ADCs are low. In this architecture, a digital multiplexer is used, unlike the previous architecture, where an analog multiplexer is used. The main advantage of utilizing a digital multiplexer is that the power-consuming buffers and ADC drivers are avoided, and the inter-channel cross-talk noise is eliminated. This is due to the fact that digital signals have high noise margins and are more stable, compared to analog signals against cross-talk and other noises. However, this architecture has higher numbers of ADCs and, consequently, consumes a higher area and power consumption. Therefore, area and power reduction techniques should be applied in the design process. This architecture is shown in [28].

In the third architecture, unlike the other two architectures in which there is one ADC for all the channels or one ADC for each channel, one ADC is shared among multiple channels. Figure 1c shows the block diagram of this architecture, where there are *m* rows and *n* columns. As shown in this figure, one ADC is assigned to each column with *m* rows through an analog multiplexer. Since these multiplexers are smaller compared to the first architecture, the design considerations to avoid cross-talk is less challenging. As another advantage of this architecture, since the number of ADCs are dependent on the number of columns (n), choosing an appropriate value for *n* results in minimum value of the power consumption and the area, especially when the total number of the channels is very high. Therefore, as a solution for the large-scale recordings, the architecture of Figure 1c is the best option. This architecture is presented in [15].

In addition to these main architectures, other non-popular architectures are reported in the literature where no ADC is involved. As an example, in [29], the analog signals are converted to time duration using the pulse width modulation (PWM) technique and transmitted to the outside of the body.

Increasing the number of channels in order to increase the spatial resolution is desirable, however, it increases the output data rate and the power consumption especially in the transmitter. In order to decrease the data rate, researchers have proposed different methods to compress the data. One of the main methods to compress the data that is utilized in neural extracellular activities is done in the time domain. In this method, only the APs are detected and transferred out of the body. Since the duty cycle of this method is in the range of 2% to 20%, the data can be compressed by a maximum factor of 50 [28,30]. This method is applicable in both digital and analog domains.

Another compression method that is used in both analog and digital domains is compressive sensing (CS). This method is simple in its implementation and efficient in compression and is suitable for neural signals, especially iEEG. In this method, instead of sending all *N* samples of the neural signal of each channel, an *M* linear combination of these samples are sent where *M* is less than *N* [31,32,33]. The compression blocks can be implemented in analog in points A1 and A2 or can be implemented digitally in points D1 and D2 in various architectures of Figure 1. In Section 6, we explore the compression methods and their challenges in more detail. In the following sections, we probe the main blocks of these architectures and their challenges.

## 4. Neural-Signal Amplifiers

As discussed in Section 2, neural signals have very small amplitude and bandwidth and are required to be amplified before converting them to digital signals by an ADC. The amplification is done in AFE with neural amplifiers. Different DC offset voltages are generated across various electrodes due to the electrochemical reaction at the electrode–tissue interface on each channel. These voltages vary from 1 mV to 50 mV [15]. The offset voltage of channel can saturate the neural amplifier due to its very high voltage value compared to the amplitude of the neural signal. Therefore, these offset voltages should be eliminated. In addition, to design multi-channel neural amplifiers for implantable applications, the power consumption and chip area should be minimized.

Different noise sources also impose challenges in the design of neural amplifiers. Neural amplifiers have very low bandwidth. Therefore, the main sources of noise are flicker and thermal noises from the neural amplifiers, which decrease the SNR in the output of the amplifiers. To achieve the adequate output SNR, the neural amplifier is designed as an LNA.

Generally, in the design of neural amplifiers, to provide the required signal quality, several factors should be considered. These factors are sufficient gain, high SNR, appropriate bandwidth, high common mode and power supply rejection ratio (CMRR and PSRR), low power consumption and low chip area.

### 4.1. Neural-Signal Amplifier Topologies

In this paper, we classify the neural amplifier topologies to AC-coupled and DC-coupled neural amplifiers based on their tissue DC offset cancellation methods. In AC-coupled neural amplifiers, the DC offsets are blocked using one or two AC-coupling capacitors placed at the input of the amplifier. To achieve high gain in the amplifier, these capacitors are usually very large. A drawback of this is that the chip area increases significantly and the input impedance decreases. A small-frequency and well-defined high-pass pole is required to record low-frequency signals while rejecting the tissue DC offset voltage.

An alternative approach to canceling the tissue DC offset effect is utilizing the DC-coupled neural amplifiers. This type of amplifier uses a low-pass filter (LPF) in the feedback path (shown in Figure 2) to generate a high-pass pole as the overall transfer function.

#### 4.1.1. AC-Coupled Neural Amplifiers

One of the most popular neural amplifier topologies is the closed-loop capacitive feedback amplifier introduced in [34], which is also known as the conventional capacitive feedback network (CFN) topology [29,35,36,37,38,39]. Figure 3a shows the conventional circuit architecture of this topology. A large capacitor $C_{I}$ at the input is used to block the tissue DC offset. The gain of this amplifier is equal to $\frac{C_{I}}{C_{F}}$. To implement the high-pass pole, a capacitor $C_{F}$ is set in parallel with a highly resistive element $R_{F}$ in the feedback path.

The main drawback of this topology is its large area due to the huge input capacitor $C_{I}$. To achieve high gain and low-cutoff frequency at the range of 1 Hz and lower, a huge capacitor at the input of the amplifier is required. This large capacitor ($C_{I}$) occupies a large chip area and results in reduced input impedance of the neural amplifier. For this reason, this structure is not suitable for multi-channel applications. Using two or three gain stages could reduce the size of the capacitors and, consequently, the chip area, at the cost of increased power consumption.

The total input-referred noise of the amplifier is presented as [34]
(1)$$\begin{array}{c} \bar{V_{ni,amp}^{2}}={\left(\frac{C_{I}+C_{F}+C_{in}}{C_{I}}\right)}^{2}.\bar{V_{ni}^{2}} \end{array}$$
where $C_{F}$ is the feedback capacitor, $C_{I}$ is the input capacitor, $C_{in}$ is the OTA input terminal capacitance, $\bar{V_{ni}^{2}}$ is the OTA input-referred noise power and $\bar{V_{ni,amp}^{2}}$ is the input-referred noise power of the whole neural amplifier.

Equation (Equation 1) shows that increasing the gain of the neural amplifier, or in other words increasing the value of $C_{I}$, for a constant $C_{F}$, reduces the overall input-referred noise (IRN). In addition, for constant gain, increasing the sizes of the differential pair transistors on the one hand decreases the IRN power of the OTA ($\bar{V_{ni}^{2}}$), and, on the other hand, increases the size of the OTA input capacitance $C_{in}$ which, according to Equation (Equation 1), results in increasing the noise multiplication factor. As a trade-off, there is an optimum point for the size of the differential pair to minimize the overall IRN power of the neural amplifier for a specific gain.

By replacing the feedback capacitor $C_{F}$ in the conventional CFN topology shown in Figure 3a to a T-capacitor network topology in Figure 3b, the total equivalent feedback capacitor is reduced [40]. Therefore, a similar gain is maintained with a smaller $C_{I}$ capacitor. However, this comes at the cost of increased low-cutoff frequency due to the reduced effective feedback capacitance. In order to maintain the same low-cutoff frequency, the feedback resistor has to increase. Increasing the feedback resistor, increases the input referred noise of the whole neural amplifier.

Another topology of the AC-coupled neural amplifier is shown in Figure 3c. In this topology, a high-pass filter (HPF) followed by a resistive feedback non-inverting amplifier is used to cancel the DC offset [41]. The HPF is composed of an electrode capacitance and a high resistive PMOS where the bias current is programmable. The operating point of $M_{P}$ in Figure 3c determines the amount of resistance and the low-cutoff frequency. Since the value of the electrode capacitance varies significantly from one electrode to the other, the low-cutoff frequency is not accurate. In [42], the same structure is used to design multi-channel AFE to construct a neural signal recording utilizing off-the-shelf components. This structure is used with an on-chip AC-coupled capacitor in the literature. However, in these designs, large off-chip resistors or active on-chip resistors implemented with MOS transistors biased in the subthreshold region are used, which increase the IRN.

Figure 3d shows an open loop network (OLN) topology. It is similar to that of Figure 3c, while utilizing an open loop amplifier [43]. This topology can consume smaller silicon area compared to CFN topology, however it suffers from non-accurate gain and also sensitivity to the technology process deviation.

Capacitive amplifier feedback network (CAFN) topology is another topology that removes the tissue voltage offset using a coupling capacitor shown in Figure 3e [44]. This structure is a band-pass amplifier where its midband gain ($A_{m}$) is calculated by $\frac{C_{2}C_{3}}{C_{1}C_{4}}$ and its low-cutoff frequency ($f_{L}$) is calculated by $\frac{C_{2}}{R_{f1}C_{1}C_{4}}$. As explained in [44] and [16], C3 has to be maximized to decrease the input referred noise. For a given gain, the ratio of $C_{3}/C_{4}$ has to increase and the ratio of $C_{2}/C_{1}$ has to decrease. By increasing $C_{3}/C_{4}$ and decreasing $C_{2}/C_{1}$, the input referred noise gets close to the CFN topology of Figure 3a. In conclusion, the added complexity to this structure does not significantly improve its parameters. Even in some cases, with the same conditions, the noise, power consumption and chip area of this neural amplifier deteriorates compared to the CFN structure of Figure 3a.

The topology shown in Figure 3f is a Miller compensated CFN (MCCFN) topology, which is similar to the conventional CFN topology, but it uses two OTAs in series. In some implementations, such as in [45,46], no Miller compensation capacitors are utilized in the OTAs. However, in other implementations, such as [47], a Miller compensation capacitor is used in the second OTA (OTA${}_{2}$) to make the non-dominant pole further away for better stability. In addition, in the design presented in [16] a Miller compensation capacitor is used in OTA${}_{1}$ to create a double pole in the low-cutoff frequency. In the design of a single stage neural amplifier with high gain, this topology can provide a fair trade off between output swing, DC gain, noise, and power consumption. This advantage is realized by utilizing the OTA${}_{2}$ in this structure, which increases the open loop gain. In addition, by designing the OTA${}_{2}$ as a high-swing OTA in this structure, we can increase the output swing. Designing neural amplifiers with this topology (MCCFN) can result in low power and low noise with a low-noise efficiency factor (NEF) compared to the other topologies. However, in advanced technologies, where the neural amplifiers are designed in multistage architecture, there is no significant advantage compared to the conventional CFN topology. Note that high-cutoff frequency in all AC-coupled topologies of Figure 3 is determined by the frequency response of OTA${}_{1}$.

#### 4.1.2. DC-Coupled Neural Amplifiers

The schematic shown in Figure 4a is the first neural amplifier that uses an LPF in the feedback path which is based on the block diagram of Figure 2 [48]. The high-cutoff frequency of this architecture is also realized by the frequency response of $OTA_{1}$ and its midband gain is obtained from the DC gain of this OTA. As a result of this feature, no large capacitor ratio is required to obtain high midband gains. However, the midband gain is affected by strong process variations.

Utilizing an integrator as an active LPF in the feedback path results in a lower area since a smaller capacitor is required due to the Miller effect compared to the structures that utilize passive LPF. However, to reduce the IRN, the capacitors $C_{I}$ and $C_{L}$ should be increased, which results in additional area. In addition, this topology consumes huge amount of power in OTA${}_{2}$ (feedback path) which in turn reduces the NEF in this topology.

A single ended configuration is utilized in Figure 4a. This topology has a lower CMRR and PSRR compared to a fully differential configuration; however, it occupies a lower area, since it only has one Miller capacitor. Moreover, the resistance of the pseudoresistor shown in Figure 4a varies due to the high voltage swing at the output and, consequently, alters the high-pass pole. The variation in the open-loop gain of the OTA${}_{1}$ can also change the high-pass pole.

A DC-coupled neural amplifier topology similar to Figure 4a is shown in Figure 4b [49]. Unlike Figure 4a, the architecture in Figure 4b utilizes a fully deferential architecture and a passive LPF. Since this architecture does not exploit the miller effect, a huge feedback capacitor is used to obtain a high-pass pole at low frequencies. Passive off-chip elements are used in the architecture of [49] that is not suitable for multi-channel and implantable applications. Since no OTA is utilized in the feedback path, the noise and NEF are reduced at the cost of a huge feedback capacitor. Note that the feedback capacitor is implemented off chip, due to the high area.

Figure 4c shows a neural amplifier that utilizes the block diagram of Figure 2 as well to remove the DC offset of the input signal [8]. However, in this method, the currents are compared in the input instead of voltages, unlike the two previous topologies. This figure shows an AC-coupled chopper stabilized instrumentation amplifier (ACCIA), which utilizes a current balancing instrumentation amplifier (CBIA) block. The coarse-fine servoloop is implemented using a coarse transconductance (CGM), a fine transconductance (FGM), and an integrator as shown in Figure 4c. In [8], the authors have minimized the output range of analog fine servo by designing the coarse servo digitally. With this technique, the power consumption of the fine servo is reduced and the power–noise performance of the ACCIA is improved when compared to the previous ACCIA reported by the same group in [50]. This design also uses a CHS technique to reduce noise. However, this design still consumes a large area and has high power consumption, which is not attractive for multi-channel applications.

The idea of utilizing a digital LPF instead of analog in the feedback path of Figure 2 in order to avoid utilizing huge capacitors and high-power OTA is proposed in [51,52] and shown in Figure 4d. In this method, the low-cutoff frequency can be determined more accurately. Also, it can result in lower area and less power consumption in advanced technologies, and a comparison between the input signal and the feedback signal is performed by utilizing a DAC and an array of transistors in a differential pair. This structure modulates and changes the width of the transistor based on the offset voltage to maintain constant IRN and CMRR. Nevertheless, measurement results show that these parameters vary with the offset voltage variation. The digital LPF in [51] is designed off chip. The off-chip implementation of the filter creates an undesired delay to the low-frequency signal path. This limits the order of the filter to first order and makes it difficult to stabilize the feedback loop. In [52], a digital LPF is implemented on chip for four channels to eliminate the delay issue. The low cut-off frequency of this design can be programmed down to 40 Hz.

The authors in [15] also exploit a digital on-chip filter to implement a 56 channels neural recording implant. In this design, comparing the input signal and the feedback signal is performed directly in the currents passing through the differential pair with a current steering I-DAC. This design uses a CHS to reduce the flicker noise, but due to the high power consumption of the blocks and relatively high bandwidth (10 Hz–5 KHz) compared to $f_{C}$ (≃100 Hz, flicker noise corner frequency), a high NEF (=7) is observed. Also, in addition to the complexity of this design, the midband gain is very sensitive to the technology process deviation.

### 4.2. Multistage Amplifiers

In the literature, there are reports of single stage amplifier designs for biomedical implants that are mostly implemented as capacitive coupled with the microelectrode sensors [53,54,55,56,57]. One-stage neural amplifiers face the following challenges. In the case of DC-coupled neural amplifiers that are generally designed as open loop structures with high gain, the midband gain is very sensitive to process variation. In addition, the gain variation causes undesired variation in the low-cutoff frequency ($f_{L}$). On the other hand, in the case of AC-coupled, very popular and commonly-used conventional CFN topology shown in Figure 3a, has the following drawbacks. The midband gain ($A_{M}$) of such an amplifier is obtained by $\frac{C_{I}}{C_{F}}$. For a specific and fixed value of $A_{M}$, the minimum possible value is chosen for $C_{F}$ to minimize the $C_{I}$ and, consequently, the chip area. The $C_{F}$ is usually chosen in the range 100–200 fF to be larger than the neighborhood parasitic capacitances. Therefore, to have a high gain, the $C_{I}$ has to be increased $A_{M}$ times greater than the $C_{F}$, which results in a huge chip area and very low input impedance. Also, the latter factor itself causes attenuation of the neural signal in the input of the amplifier, which in turn reduces the total gain of the amplifier and the CMRR. Furthermore, in the scaled and advanced technologies, the transconductance ($g_{m}$) reduces due to short-channel effects of MOS transistors, which creates challenges and difficulties in designing high gain OTAs.

To overcome these challenges, amplifiers are designed in two or three stages in the AFE of the neural recording systems. Utilizing AC-coupled multistage amplifiers reduces the input capacitors’ value that leads to a reduction in the chip area at the expense of slightly increasing the power consumption. A few examples of such multistage amplifiers are provided in [58,59,60,61,62].

In order to have low IRN, the first stage is designed as a low noise amplifier (LNA). The other stages (i.e., the second and third stages) are designed to provide enough gain and linearity. Also, tuning the gain, as well as the $f_{L}$ and $f_{H}$, is carried out in the second or third stages. Although it may be observed in [37] that $f_{L}$ and $f_{H}$ are tuned in the first stage of the neural amplifier, this is not desirable as it affects the IRN.

It has been proven by *Isoperimetric Theorem* in mathematics that for two or three-stage AC-coupled amplifiers (assuming that the main area of the neural amplifier is consumed by the capacitors), the maximum total gain for a specific area or the minimum area for a specific total gain is obtained when the gains at all stages are equal. However, further design considerations, such as lowering the IRN, causes the first stage of the neural amplifier, which is an LNA, to be designed with higher gain compared to the other stages [37,63].

Another challenge seen in both single-stage and multistage amplifiers (in the second or third stage) is that the high output swing of the OTA varies the feedback resistance of the MOS pseudoresistor in the CFN topology of Figure 3a. This variation increases the non-linearity of the amplifier and consequently increases the distortion and makes the high-pass pole frequency variable with time [64].

Figure 5a illustrates a solution to this problem that is exploited in [65,66,67]. In these designs, the front-end amplifiers are composed of two stages, in which the second stage utilizes a source follower such as that in Figure 5a to provide a constant voltage for the gate-source terminals of the MOS pseudoresistors, while the output swing is high. This technique increases the linearity of the amplifier significantly.

Nowadays, many designed and proposed front-end amplifiers have two or three stages. The design in Figure 5b utilizes of two stages [65,66]. The gain of the first and second stages are as 39 and 14, respectively. To increase the linearity in the first stage, NMOS transistors are used as pseudoresistors and in the second stage as shown in this figure, NMOS and PMOS transistors accompanied by a source follower, are used. The pseudoresistors are all thick-oxide transistors, which provide higher resistance compared to the standard CMOS transistors. The high-pass pole frequency is adjusted with the bias current of the source followers in the second stage. This happens while, the low-pass pole frequency is determined by $C_{L}$. This extra and large capacitor is used in [34,36,68], which, in all of these designs, causes an overhead in the consuming area.

The architecture shown in Figure 6a utilizes three stages for amplification [63]. The first stage is an LNA, the second stage is band-pass filter where the low and high-cutoff frequencies are adjusted. The third stage behaves like a variable gain amplifier (VGA) and a buffer. The gains of the first, second, and third stages are 50, 2, and 5, respectively. As illustrated in Figure 6b, the second stage utilizes a current-controlled pseudoresistor, that uses a cross-coupled architecture to provide a symmetrical resistance with high linearity in the range of 0.2 V. This voltage is at the voltage swing range of the second stage. The low-cutoff frequency of these circuits is determined by the bias current of the current sources, while altering the value of $C_{L2}$ tunes the high-cutoff frequency.

### 4.3. Noise Reduction Techniques

As illustrated in Table 1, the amplitude of the neural signals are very low; therefore, to achieve a high SNR, the first stage of a multistage neural amplifier should be an LNA. Since the noise in the second and third stages of the amplifiers is divided by the squared gain of previous stages, the input referred noise of these stages is less important [69]. The noise efficiency factor (NEF) is a widely used figure of merit and is presented as [70]
(2)$$NEF=V_{ni,rms}\sqrt{\frac{2I_{tot}}{\pi .U_{T}.4kT.BW}}$$
where $V_{ni,rms}$ is the input referred rms noise voltage, $I_{tot}$ is the total supply current of the amplifier, and BW is the amplifier bandwidth (in Hz). To compare LNAs with a different IRN, power consumption and bandwidth, NEF is utilized and the smaller NEF is better.

To reduce the noise of LNAs, we explain circuit and systematic approaches in the following subsections. Both of these approaches are applicable to AC and DC-coupled neural amplifiers. IRN and NEF in different topologies are calculated and compared together in [16]. However, since the conventional CFN topology of Figure 3a is more appropriate and popular for multi-channel neural recordings compared to other topologies, we focus more on this topology in this paper.

#### 4.3.1. Circuit Techniques

The IRN value of an OTA varies based on its architecture. For example, a two-stage OTA (or Miller OTA) could have a lower IRN compared to folded cascade architecture, due to the lower number of transistors in the first stage. However, in general and in the same condition of bias current and transistor sizes, the differential pair transistors have a maximum contribution in the IRN value of various OTA architectures. Also, the transistor of the tail current source (the current source that is connected to the differential pair) in all architectures as well as in the cascode transistors in the telescopic or folded cascode architectures has the minimum contribution in the IRN value.

As presented in Table 1, neural signals have very low frequency components and small band width. For the frequencies higher than the corner frequency $\left(f_{c}\right)$ in an LNA, the thermal noise is dominant and for the frequencies lower than $\left(f_{c}\right)$, the flicker noise is dominant, where both of these noises should be mitigated. For example, in [34] and [48], $\left(f_{c}\right)$ is reported as 100 and 300 Hz, respectively. In [14], the IRN of the thermal noise is calculated and tabulated for different OTA architectures. Equation (Equation 3), presents the flicker noise power of a MOS transistor [71].
(3)$$\begin{array}{c} \bar{V_{n}^{2}}=\frac{K}{C_{ox}WL}.\frac{1}{f} \end{array}$$
where $C_{ox}$ is the gate oxide capacitance per unit area, and K is a process dependent constant for a MOS transistor. W and L are the width and length of a MOS transistor, respectively. To reduce the flicker noise power of an OTA, the differential pair transistors should be a PMOS type, which have lower K compared to the NMOS transistors and according to Equation (Equation 3), the size chosen should be large.

In addition, based on the IRN equations of the thermal and flicker noise in various OTA architectures, increasing the transconductance of the differential pair $\left(g_{m}\right)$ compared to other transistors reduces the IRN that corresponds to both thermal and flicker noise. Therefore, by biasing the differential pair in the subthreshold region, the $g_{m}$ of these transistors could be maximized. This can be achieved by increasing the W/L ratio of the differential pair for a constant bias current. Increasing the W/L ratio also decreases the flicker noise of the differential pair.

As explained in Section 4.1.1, for the CFN topology, increasing the $C_{I}$ based on Equation (Equation 1), increases the midband gain of the neural amplifier and results in a reduction of the IRN. This is why the gain of the first stage (LNA) is usually designed to be significantly higher than the following stages. Also, for a given gain, increasing the width of the differential pair, decreases the flicker noise power of the OTA in Equation (Equation 1) on one hand, and increases the $C_{in}$ on the other hand. Therefore, as a trade off, there is an optimum point for the W where IRN is minimized. The low noise neural amplifier in [34,63,65,72] has been designed to consider the circuit noise reduction techniques explained in this section, but without considering the systematic technique of the following section.

#### 4.3.2. Systematic Technique

The neural signal contains important information in the low frequency range (i.e., less than $f_{c}$). In this frequency range, the flicker noise is dominant and, to reduce its effects, the chopper stabilization (CHS) technique is usually used [73,74]. The CHS technique operates as follows.

Utilizing up-modulation by the first chopper, CHS transposes the neural signal to a higher chopping frequency ($f_{ch}$) where the 1/f noise is not available. In the next step, the amplifier amplifies the signal while adding its 1/f noise and offset. The second chopping modulates up the offset and 1/f noise to $f_{ch}$ while the signal is demodulated back to the baseband. A LPF extracts the original signal with a much higher SNR.

CHS can be used in both AC and DC-coupled amplifiers. Examples of utilizing this technique for DC-coupled amplifiers are shown in [8,15,50]. However, in this section, we will further explore the CNF topology of Figure 3a. This is because this topology is more appropriate for multi-channel neural recordings and is more commonly used.

In practice, implementing the CHS technique is challenging. The fact that the chopping switches are not ideal, as well as the amplifier, which has offset, limited gain and bandwidth, creates challenges in the design process. Utilizing the CHS technique in a designed LNA with the CFN topology of Figure 3a for the same capacitor size does not reduce the noise at low frequencies, but rather it increases it. The reason for this is explained in [75]. Figure 7a shows the amplifier utilized in [75] with the input and output chopper switches and the IRN of OTA. The input chopper switches, accompanied by the OTA input parasitic capacitance (such as the switched capacitor circuits), can be modeled as a resistor, with a value that can be calculated as
(4)$$\begin{array}{c} R_{eq}=\frac{1}{f_{ch}C_{in}} \end{array}$$
where $C_{in}$ is the OTA input parasitic capacitance and $f_{ch}$ is the chopping frequency. The total noise of the neural amplifier, when transferring the OTA noise to the input of the neural amplifier by considering the effect of the resistor and other capacitors, could be presented as
(5)$$\begin{array}{c} \bar{V_{in}^{2}}=\bar{V_{in,OTA}^{2}}{\left(1+\frac{C_{F}}{C_{I}}+\frac{2\pi f_{ch}C_{in}}{sC_{I}}\right)}^{2} \end{array}$$

This equation shows that reducing the total IRN of the amplifier requires a huge $C_{I}$ capacitor. The capacitance value of the $C_{I}$ in [75] and [76] are reported as 300 pF and 1 nF, respectively. These large capacitors increase the area and also decrease the input impedance of the neural amplifier. The latter factor reduces the CMRR and also increases the effect of the main interferences on the system [77]. To improve the input impedance of this neural amplifier, an input impedance boosting circuit is utilized in [75,76] as shown in Figure 7b. The operation of this circuit is such that a larger portion of the input capacitor current is provided by the output of the OTA through positive feedback. This technique increases the input impedance in [75] from 400 MΩ to 2 GΩ at 1 Hz.

Two other challenges regarding the chopper amplifiers are reducing the output ripple and their residual offsets [75,78]. The offset voltage of the OTA causes the ripple. When the offset voltage becomes amplified after up modulation, it can even have an amplitude that is greater than biopotential signals; therefore, it can limit the amplifier’s output headroom. The amplifier designed in [78] senses the ripple in the output using a ripple reduction loop (RRL), and compensates for the ripple by applying it to the input through a feedback. Note that since RRL is implemented in analog and works continuously, it increases the power consumption significantly.

The residual output offset is mainly due to the non-idealities of the CMOS switches in the input chopper [73]. Clock feedthrough and charge injection creates spikes in the input of the OTA and they are amplified and presented in the output of the OTA. Then, these spikes are down modulated with the output chopper and increase the DC at the output, which is called the residual output offset. The authors in [76] have compensated for the residual offset by embedding the circuit, called DC servo loop (DSL) in the feedback path. DSL takes samples from the output and provides almost equal current, but in the opposite direction to the offset current by utilizing a $G_{m}-C$ filter and applying it to the input of the chopper. The main drawback of this circuit is that it requires a huge off-chip capacitor (greater than 10 μF) to decrease the low-cutoff frequency to the appropriate value.

To reduce the power consumption while removing the ripples in [78] and to avoid huge off-chip capacitor in [76], a new mechanism called a digitally-assisted calibration loops is proposed in [75] that removes both the ripple and the offset. This mechanism provides a digital implementation of both RRL and DSL. There are also more examples in the literature that utilize techniques to reduce the charge injection [79]. Three examples of these techniques that are employed in the chopper amplifiers are nested chopping, spike filtering, and the use of delayed modulation or a dead-band.

Figure 8a shows the schematic of the nested chopper amplifier [80,81,82,83]. As illustrated in the figure, this architecture utilizes a pair of internal and external choppers, where the frequency of the internal chopper should be chosen to be greater than 1/f noise corner frequency ($f_{C}$). The frequency of the external chopper ($f_{chL}$), which is a relatively low frequency, can be optimized for the input signal efficiently. With this technique, [80] has achieved 100 nV offset, with $f_{chH}=2$ KHz and $f_{chL}=15.6$ HZ.

A spike filtering technique is another approach for reducing the residual offset. As explained earlier, the charge injection in the MOS switches of the input chopper creates spikes in the output of the amplifier, which eventually appears in the output of the second chopper as the residual offset in the frequency spectrum. Therefore, we can assume that these spikes have odd harmonics in the frequency domain [84]. By filtering these spikes, the residual offset can be reduced significantly. Figure 8b shows the schematic of this technique used in [85,86].

Unlike the previous method, where a filter is used to reduce the residual offset in the frequency domain, in the delayed modulation or dead-band technique, the residual offset is mitigated in the time domain. Figure 9a shows the schematic of the delayed modulation technique. As shown in the figure, this technique utilizes a modulator M, amplifier $A_{1}$, and demodulator D. In [87], an amplifier is used followed by a LPF to shape the spikes and the second chopper with a constant delay is used (as shown in Figure 9b) to demodulate the spikes. This causes the average of the signal to be zero which is extracted by a LPF. Another method, which is called dead-band or guard-band, that has a relatively simpler implementation, is shown in Figure 9c. In this method, there is no modulation when there is spike in the output signal. This technique is used in [88,89].

Another approach to implementing the CHS technique in the AC-coupled CFN topology, as shown in Figure 10a, is to use a chopper in front of the input coupling capacitors [60,90]. In this figure, there are two feedback loops that determine the midband gain and high pass corner of the amplifier. However the main drawback of this architecture is its low input impedance, which is calculated as
(6)$$\begin{array}{c} Z_{in}=\frac{1}{j\omega_{sig}\left(1+f_{ch}/f_{sig}\right)C_{I}} \end{array}$$
where $f_{ch}$ is the chopping frequency and $f_{sig}$ is signal frequency. Increasing the $f_{ch}$ causes a more relaxed implementation of LPF on the one hand, and on the other hand, it reduces the input impedance according to Equation (Equation 6). The input impedance is reported to be about 8 MΩ in [60]. In order to improve the low impedance of the design in [60,90], a positive feedback loop (PFL) shown in Figure 10b is proposed in [91]. This circuit takes samples from the output and then provides a large portion of the current of the input coupling capacitors. The measurement results show that the input impedance increases from 6 MΩ to 30 MΩ. Note that due to the presence of parasitic input capacitances, the boost factor and the original input-impedance are lower than the expected (100 and 8 MΩ, respectively).

Finally, at the end of this section, we analyze [15], as a sample of the DC-coupled neural amplifier. Figure 11 shows the block diagram of this amplifier, which has 56 channels. In this architecture, a digital DSL is used in each channel to remove the input DC offset (caused by recording electrodes). In addition, CHS is used to reduce the 1/f noise of the LNA. Avoiding huge capacitors in the DC coupled architecture due to utilizing a digital DSL reduces the chip area. However, due to utilizing CHS at a much higher bandwidth (10 Hz–5 KHz) than $f_{C}$ (≃100 Hz, flicker noise corner frequency), it does not present a good noise reduction performance. The reported IRN and NEF are 5.4 $\mu V_{rms}$ and 7, respectively, with a offset voltage of 50 mV.

### 4.4. Advanced Neural-signal Amplifiers

In this section, we compare and summarize the various mentioned topologies of neural amplifiers for multichannel neural recording application in Table 2. Also we investigate the noise reduction techniques in advanced technologies. DC-coupled neural amplifier is not appropriate for large-scale recording application due to following drawbacks. The high gain value of this amplifier has much variation due to its open loop implementation. Also it needs a huge capacitor in the feedback path where it implements a passive analog integrator and dissipates large power where it implements an active analog integrator. In case of digital implementation of the integrator for each channel, it consumes relatively large silicon area and power consumption. The best choice is to design the neural amplifier in two or even three gain stages with AC-coupled CFN topology to obtain the necessary gain in lower area. The gain of the first stage should be higher compared to the other stages to decrease the IRN of the whole amplifier. If achieving the required SNR in the output of the amplifier is possible by applying the circuit noise reduction techniques, it is not necessary to utilize the systematic techniques due to its silicon area and power overhead. Otherwise, the first stage of the neural amplifier as an LNA needs CHS technique to reduce the flicker noise of the LNA. Applying CHS to decrease the IRN needs to increase the input capacitors value which causes the input impedance reduction and consequently, CMRR reduction. Input impedance boosting circuit increases the input impedance by utilizing a positive feedback as mentioned in Section 4.3.2. Furthermore, offset voltage of OTA causes ripple at the output of the amplifier which can be reduced by the RRL circuit. Also, the non-idealities of the first chopper switches (charge injection and clock feedthrough) cause the residual offset which can be compensated by DSL circuit in the feedback path. However, we presented three methods to decrease the residual offset at the origin in Section 4.3.2. Among these methods, we suggest the nested chopper technique (Figure 8a), because of its simple implementation and high performance.

## 5. Analog to Digital Converters

The amplified neural signals are converted to digital by an ADC as shown in Figure 1, before transmitting it out of the body. This is because the digital data is very tolerant of noise and other interferers, compared to analog signals. As the neural signals have very low bandwidth and require low power consumption for implant applications, successive approximation architecture analog to digital converters (SAR ADCs) are one of the best options. This ADC utilizes digital to analog converter (DAC) and successive approximation register (SAR) blocks. Most of the neural recording systems utilize an 8 to 10 bit resolution SAR ADC [92], however in [93], the resolution of the ADC is adaptively configured by the activity of the input neural signal, to save the power and compress the data. Figure 12 shows a SAR ADC which uses charge redistribution between the capacitors. Also, two sub DACs are utilized to decrease the total amount of capacitance and consequently reduces the total area [15,65,94]. This ADC is suitable for neural recording architectures shown in Figure 1a,c which use one or several ADCs for all of the channels. Due to their large area and high-power consumption, this ADC is not suitable for the architecture shown in Figure 1b where each channel uses one ADC. For this purpose, the authors in [95] propose an improved SAR ADC shown in Figure 13a. The output bits of this ADC are extracted in serial. In addition, this structure exploits binary search algorithm as shown in Figure 13b.

Although, in general, SAR ADC is a common architecture used in neural recording implants, the use of other ADC architecture is sometimes presented in the literature. For example, a logarithmic pipeline ADC with 8 bit resolution is used in [39]. The logarithmic encoding is used to represent the high dynamic range with a short word length. Another neural recording system with 256 channels is shown in [96]. In order to reduce the area and power consumption, an 8 bit resolution single slope ADC is used for each channel. The ramp generator and counter are shared between all channels in this architecture. The authors in [97] introduce a 10-bit resolution dual mode SAR and single-slope ADC for neural recording application. In the normal mode, the ADC works as a SAR ADC to quantize the extracellular action potential. In the compression mode, to reduce the dynamic power, the ADC is configured to single-slope and it processes just essential parts of spike waveforms.

Sigma-Delta modulators are also utilized in neural recording applications due to the low-frequency bandwidth of neural signals [98,99]. Sigma-Delta modulators operate based on oversampling data conversion. Therefore, although this data converters can be low power, they can increase the power consumption in the subsequent circuits especially in the wireless transmitter due to increasing the output data rate of ΣΔ modulators much higher than Nyquist-rate ADCs. However, as mentioned earlier, the charge redistributed SAR ADC is the best data converter option for multi-channel neural recording implants.

## 6. Data Compression

Increasing the number of channels improves spatial resolution on the one hand, and on the other hand, increases the output data transfer rate and power consumption, especially in the wireless transmitter. One of the methods to decrease the power consumption is to decrease the data transfer rate by reducing the data redundancy, which is called data compression. Since the encoders of the compression methods are implanted on the brain, the method that has a simpler encoder in terms of consuming less area and power is desirable.

As mentioned earlier in Section 3, CS is a compressing technique that efficiently acquires and reconstructs a bio-signal, especially EEG and iEEG signals. CS is only used for bio-signals that are sparse in time or other domains. Luckily, most of the bio-signals are sparse in the time, Gabor or wavelet domains [100,101], which makes them suitable to use CS. Data reconstruction can be achieved with far fewer samples compared to the Shannon–Nyquist sampling theorem by utilizing optimization methods and by exploiting the sparsity of the signal.

The core of CS encodes an N-dimensional sampled input signal (X) into an M-dimensional sequence of measurement (Y) through a linear transformation by the M×N measurement matrix Φ, where Y=ΦX. In this matrix equation, M is less than N (M<N) that represents the compressing of data from N sample to M sequence. There are an infinite number of feasible solutions for X, as the equation is underdetermined. Assuming X is sparse, the sparsest solution is often the correct solution with high probability. As presented in [102], employing a random measurement matrix Φ, as a universal encoder as well as large enough input samples X is required to perform signal reconstruction of any sparse signal. A general approach to facilitating an efficient circuit implementation of Φ is to utilize a pseudo-random Bernoulli matrix where each entry $\phi_{m,n}$ is ±1 [103,104].

As shown in Figure 14, a CS encoder can be implemented in both an analog and a digital domain. Figure 14a demonstrates the block diagram of an analog implementation of a CS encoder presented in [72,94,105,106]. The sparsity of the EEG signal in the Gabor domain is utilized in [94] and the design in [72,105,106] exploits the spatial sparsity of the iEEG signals recorded from the electrodes of the sensor array. The CS core block diagram in the digital domain is shown in Figure 14b. The sparsity of the neural signal in the Gabor domain is utilized in [107,108,109,110,111,112] and implements the CS encoder digitally.

Another signal compression method, which is suitable for extracellular recordings, is presented in the literature [28,30]. This compression method is based on the sparsity of APs in the time domain. Since most of the information corresponding to the extracellular activities that are sensed and captured by the microelectrodes are in the APs, and the duty cycle of the APs is between 2% and 20%, it is sufficient to detect only the APs and transfer them out of the body [14]. As the waveform of the APs captured by different neurons in a microelectrode are different, we can exploit the features of these waveforms for the subsequent processes such as spike sorting. These waveform features could be for the time of occurrence and the maximum and minimum amplitude value. Note that the best performance is achieved when complete waveform representations are available [113,114].

An analog compression block is usually placed after the neural amplifier. Therefore, the neural signals have a relatively high voltage amplitude. One of the simplest methods for detecting and extracting the location of the APs is using a comparator in order to compare the neural signals with a threshold voltage [115,116]. The APs are detected while the neural signal cross the voltage threshold. The implementation of the detection circuit is simple in both the analog and digital circuits. However, in this method, we can only capture a portion of the AP that is above the threshold and the rest are neglected. Furthermore, accurate detection of this method is only feasible for high SNR [117]. To optimize the detection rate, the threshold voltage is chosen very carefully based on the level of the noise in the channel. This noise consists of background neural noise, flicker noise and thermal noise. Another similar method which is more effective in raising the detection rate is based on exploiting a bilateral threshold to consider both positive and negative signal polarities as presented in [28,54,118,119,120]. Saving a signal in a buffer and transmitting it with a very short delay allows for the whole waveshape of the APs to be captured, without missing any portion of them. Employing an SRAM as a data buffering block and the bilateral threshold technique in [28] improves the accurate detection rate.

In order to improve the SNR and, consequently, to increase the accurate detection rate of the APs, a pre-processor block is used in the designs in [121,122]. In this method, on one hand the waveform of the neural activities is emphasized and on the other hand the noise is attenuated to increase the SNR. This method also helps to choose the threshold voltage easily to detect the APs accurately. In practice, the implementation methods of the pre-processor in the literature are different. For instance, in [121], a pre-processor detector and spike sorting system are presented, which operates based on the variance of the neural signal. Another example is presented in [122], where an energy-based pre-processor is implemented utilizing low-power current-mode circuits.

Adaptive threshold is also another method to maximize the detection rate [123,124,125]. In this method, the threshold of detection is not a constant value and varies dynamically based on the SNR of the neural signal and the background noise. Moreover, compression and other processes such as spike detection, feature extraction, and spike sorting can be implemented digitally in digital signal processors (DSPs). Refs. [126,127,128,129] provide samples that exploit DSPs to carry out such processes.

At the end of this section, it is necessary to mention that the application of these methods are based on the input signals. The CS method is the appropriate method for the EEG and iEEG signals due to its simplicity of implementation and no need to high SNR. However, for the extracellular neural activities which is important to extract the spikes for subsequent processes such as spike sorting, the data compression method based on the threshold is suitable and the adaptive threshold method is proposed.

## 7. Conclusions

In this paper, we briefly explain the necessity of the neural recording, especially the invasive method of implanting a chip on the brain in the skull. The neural signals and their electrical specifications are also discussed. The large-scale channel recording requirements, as well as utilizing the advanced fabrication process, have imposed new challenges in the design of neural recording implants. The two most important parameters that should be considered in these designs are power consumption and silicon area. We review the various architectures of neural recording systems and conclude that architecture that utilizes an ADC for each column is the best option for the very large-scale recording. We provide detailed explanations of each block of these architectures. The most challenging block of a neural recording implant is the neural amplifier. Therefore, this block is elaborated in terms of designing a compact, high gain, low power, and low noise amplifier. We demonstrate several typologies for both AC and DC-coupled neural amplifiers. Employing a multistage amplifier to obtain high gain and to reduce the chip area for an AC-coupled neural amplifier is described. Also, various techniques to reduce the noise of the neural amplifier are discussed in detail. Although SAR ADC is the best choice for neural implant applications, we provide a complete survey of all other ADCs presented in the literature. In the end, we evaluate the data compression methods to decrease the output data rate and power consumption.

## Figures and Tables

**Figure 1 sensors-20-00904-f001:**
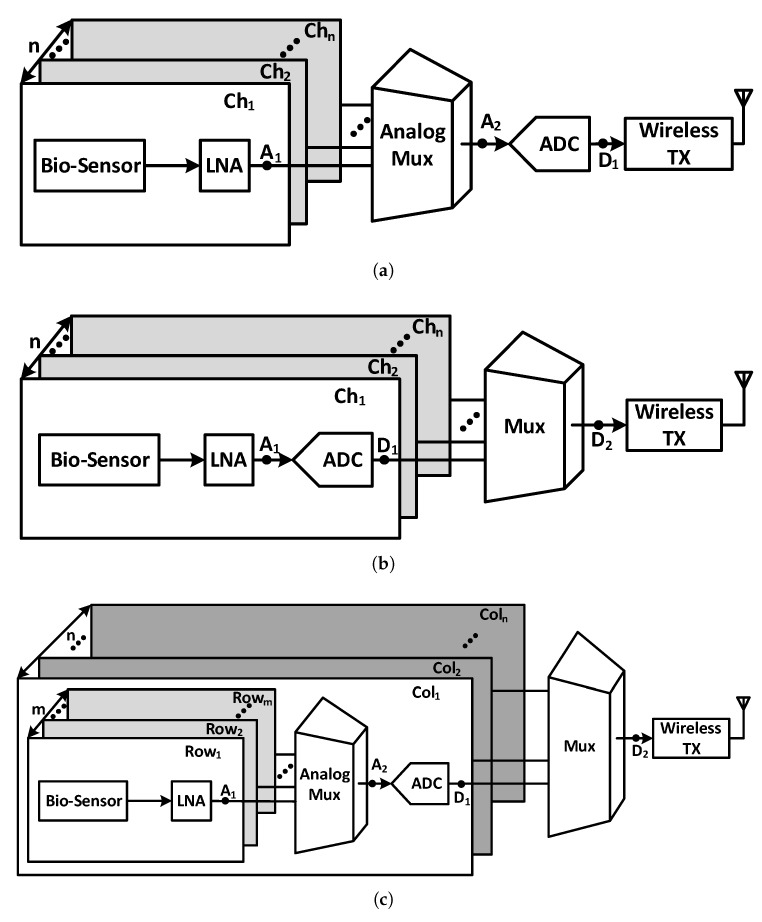
Block diagram of different multi-channel neural recording architectures. (**a**) This architecture shares an analog to digital converter (ADC) among all of the channels. (**b**) This architecture utilizes an ADC for each channel. (**c**) This architecture shares an ADC at each column.

**Figure 2 sensors-20-00904-f002:**
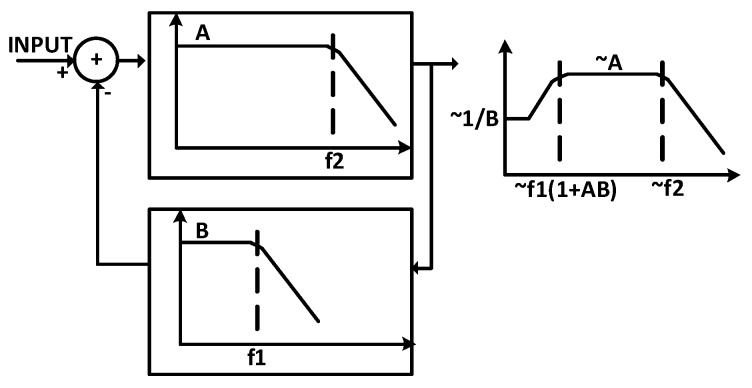
Implementing a high-pass pole using a low-pass filter in the feedback.

**Figure 3 sensors-20-00904-f003:**
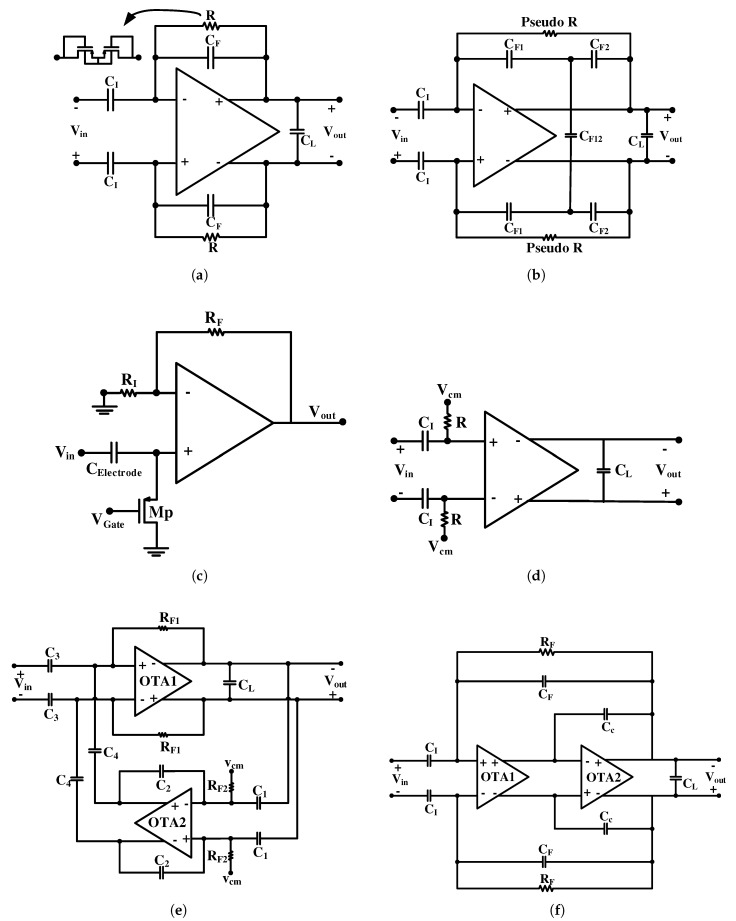
AC-coupled neural amplifier topologies. (**a**) Conventional capacitive-feedback network (CFN) topology. (**b**) CFN amplifier using T-capacitor feedback network topology. (**c**) AC-coupling utilizing the electrode capacitance and a resistive feedback. (**d**) Open loop network (OLN) topology. (**e**) Capacitive amplifier feedback network (CAFN) topology. (**f**) Miller compensated CFN (MCCFN) topology.

**Figure 4 sensors-20-00904-f004:**
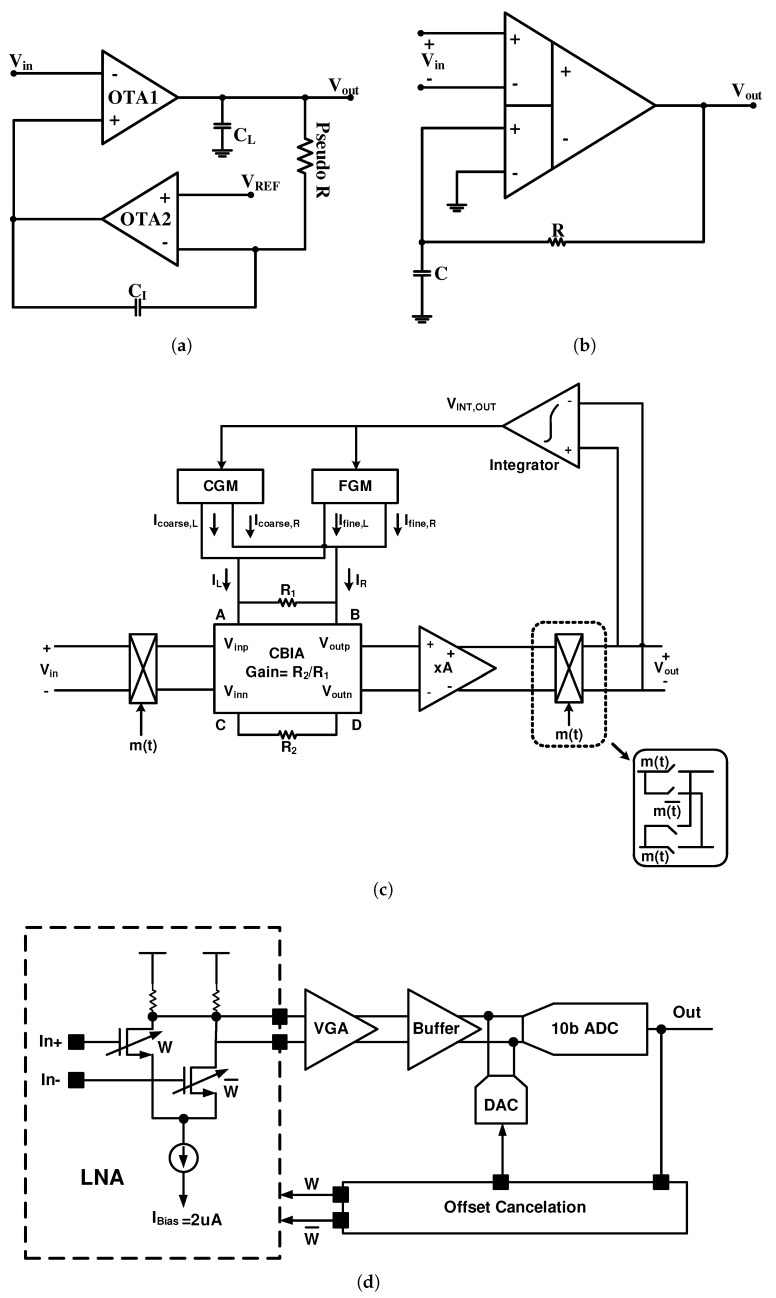
DC-coupled neural amplifiers utilizing (**a**) an analog active low-pass filter (LPF); (**b**) an analog passive LPF and a differential difference amplifier; (**c**) analog and digital DC servo loops (DSL); and (**d**) input differential pair width modulation by an offset cancellation feedback.

**Figure 5 sensors-20-00904-f005:**
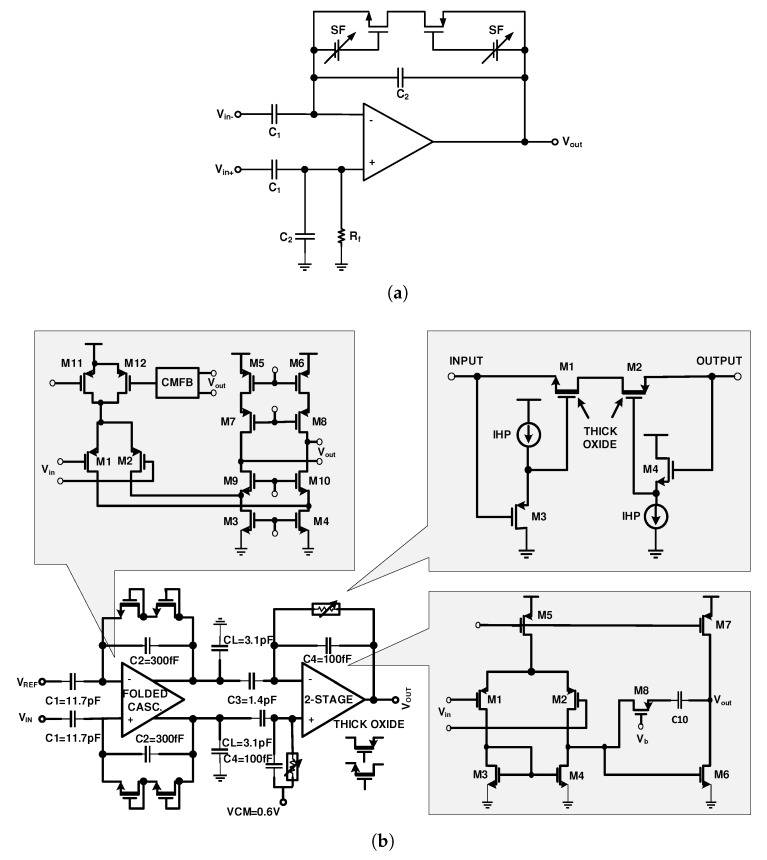
Improving the linearity of a CFN amplifier utilizing (**a**) source-followers (SF). (**b**) Two gain stages employing the proper NMOS and PMOS pseudoresistor and SFs.

**Figure 6 sensors-20-00904-f006:**
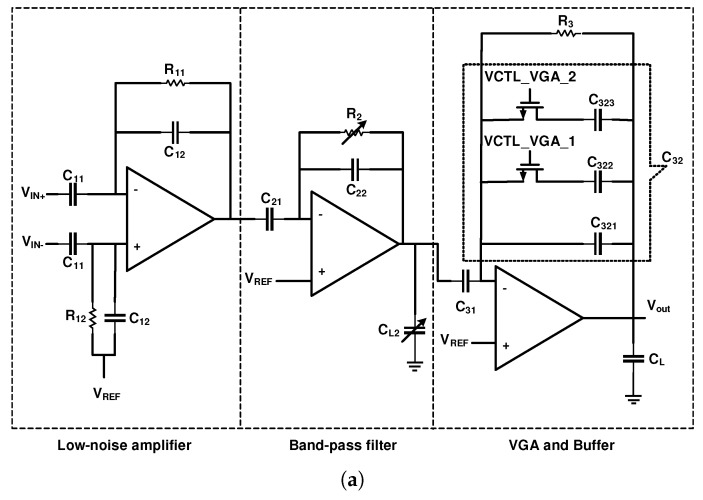
(**a**) Schematic of a three-stage amplifier. (**b**) Second-stage amplifier tunes the high and low-cutoff frequency.

**Figure 7 sensors-20-00904-f007:**
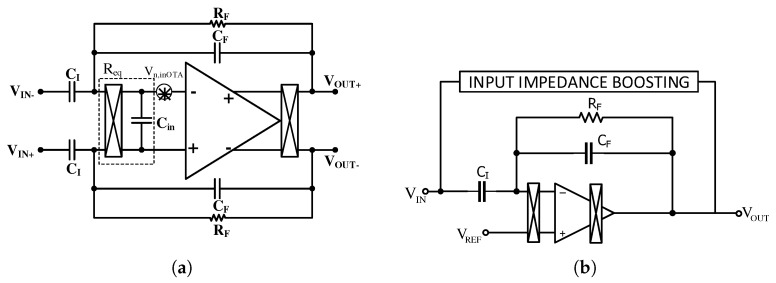
Chopper stabilization (CHS) technique in the CFN neural amplifiers. (**a**) Noise model of a neural amplifier when the first chopper is placed in front of the OTA; (**b**) Utilizing an impedance boosting feedback circuit to increase the amplifier input impedance.

**Figure 8 sensors-20-00904-f008:**
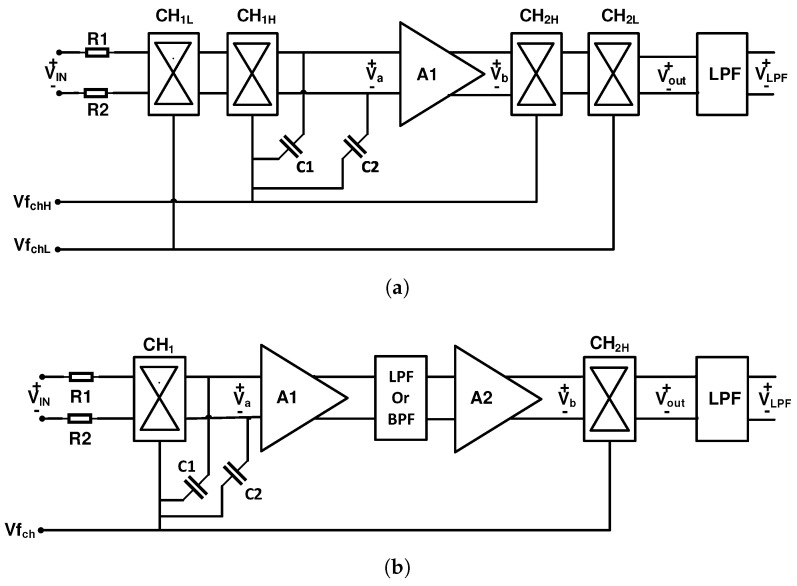
(**a**) Schematic of the nested chopper amplifier technique; (**b**) Schematic of the Spike filtering technique.

**Figure 9 sensors-20-00904-f009:**
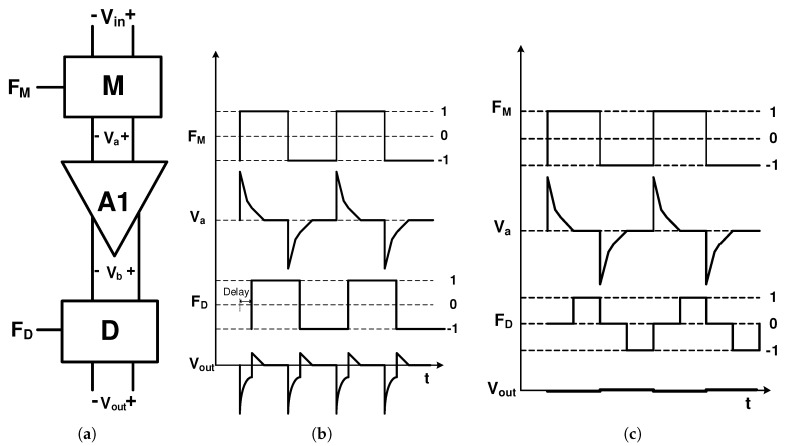
(**a**) Delayed modulation technique; (**b**) delayed demodulation clock diagram; and (**c**) dead-band clock diagram.

**Figure 10 sensors-20-00904-f010:**
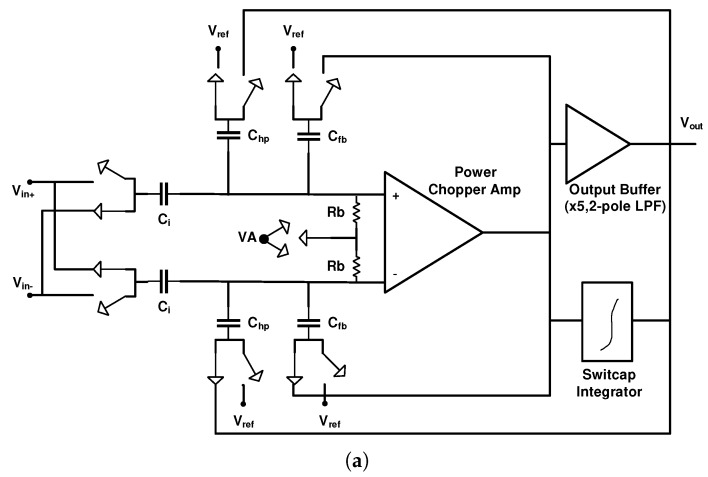
Chopper switches placed in front of the AC-coupling capacitors. (**a**) Circuit architecture of a neural amplifier, illustrating two loop feedback paths to define the midband gain and high pass corner; (**b**) schematic of the capacitively-coupled low noise amplifier (LNA) with the positive feedback loop for input impedance boosting.

**Figure 11 sensors-20-00904-f011:**
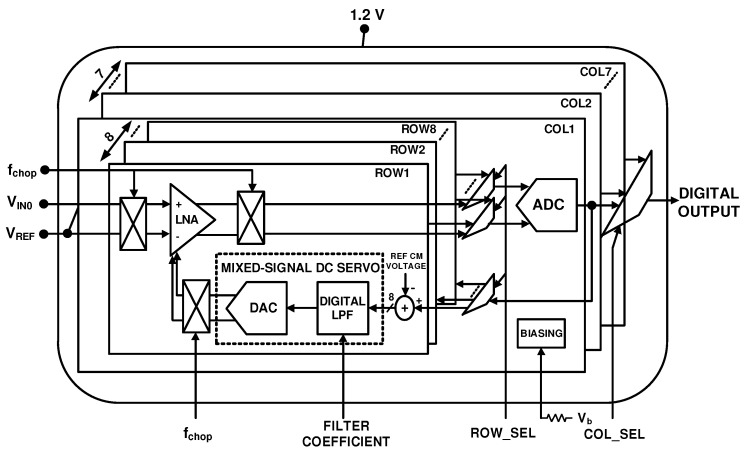
A block diagram of the 56-channel neural recording implant utilizing DC-coupled FEA and digital DC servo loop (DSL).

**Figure 12 sensors-20-00904-f012:**
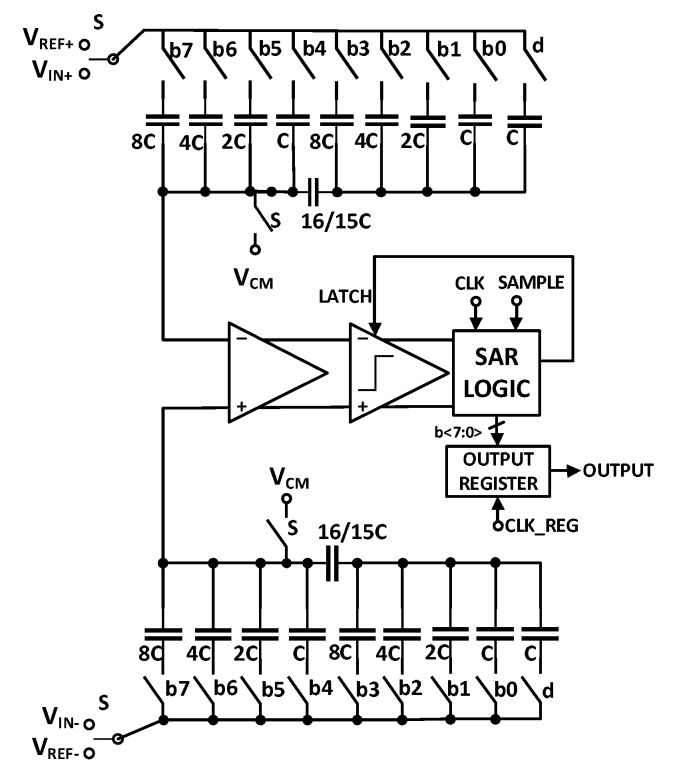
Differential 8-bit SAR ADC.

**Figure 13 sensors-20-00904-f013:**
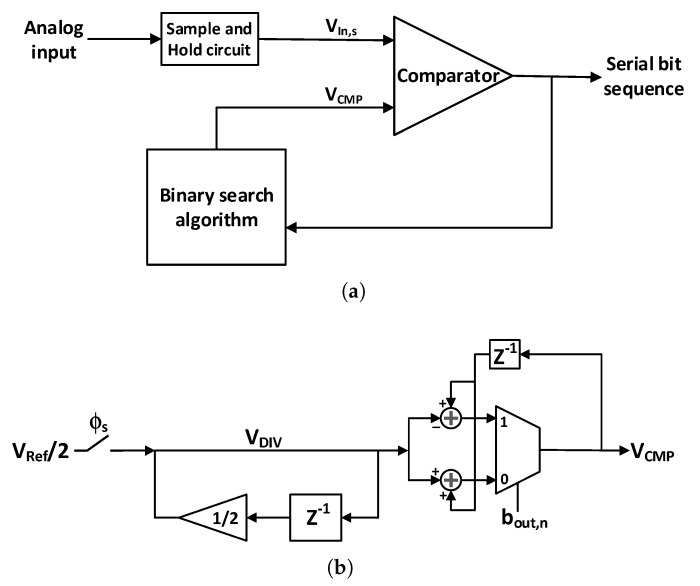
Modified successive approximation architecture analog to digital converters (SAR ADCs) for single-channel application. (**a**) Block diagram of the modified SAR ADC; (**b**) Schematic of binary search algorithm.

**Figure 14 sensors-20-00904-f014:**
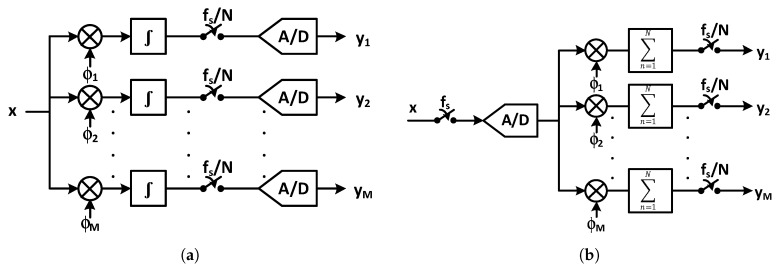
Block diagram of compressive sensing (CS) core of (**a**) an analog implantation (**b**) a digital implantation.

**Table 1 sensors-20-00904-t001:** Neural signal parameters.

Signal Type	Amplitude	Bandwidth
Extracellular action potential	50–500 μVpp	100 Hz–10 kHz
Intracellular action potential	10–70 mVpp	100 Hz–10 kHz
Local field potentials	0.5–5 mVpp	1 mHz–200 Hz
Electroencephalogram	10–400 μVpp	1 mHz–200 Hz
Electrocorticography	10–1000 μVpp	1 mHz–200 Hz

**Table 2 sensors-20-00904-t002:** Comparison between different topologies of neural amplifiers.

Amplifier Topology	Figure	Ref.	Pros	Cons
CFN	3a	[34]	Accurate gain, suitable for multistage amplifier	Large cap. area to obtain high gain
CFN with T-network	3b	[40]	Input and total cap. reduction	Low-cutoff frequency increase
Electrode cap. and resistive feedback	3c	[41]	No need to input cap.	Inaccurate and not adjustable high pass pole
OLN	3d	[43]	Small input cap. area	Inaccurate gain
CAFN	3e	[16,44]	Smaller total cap. area compared to CFN	Higher power consumption and noise compared to CFN
MCCFN	3f	[45,46]	Higher swing as a single stage high gain amplifier compared to CFN	Higher power consumption compared to CFN, Higher area consumption compared to multistage CFN
Analog Integrator	4a	[48]	Elimination of input cap.	Inaccurate gain and Low-cutoff frequency, Large power consumption
Differential difference amplifier	4b	[49]	Elimination of input cap.	Inaccurate gain and Low-cutoff frequency, large off-chip passive components
Analog & digital DSL	4c	[8,50]	Relax analog DSL requirement due to digital DSL	Large area and power consumption
Differential pair width modulation	4d	[51,52]	Fully-digital DC offset rejection	IRN and CMRR variation with input offset variation, complexity overhead
Fully-digital DSL	11	[15]	Fully-digital DC offset rejection	Inaccurate gain, high power consumption and NEF

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
