# Peer review of "Multi-Channel Neural Recording Implants: A Review"

_sensors, 2020, doi:10.3390/s20030904_

Round 1

Reviewer 1 Report

The authors present an elegant review on multi-channel neural recording Implants, addressing all the major aspects in a comprehensive approach. The paper reads well, is very well supported, requiring only minor spelling corrections. I congratulate the team for the good effort.

Reviewer 2 Report

In this paper, authors provide a review of multi-channel neural recording implants. After presenting various neural-signal features, they investigate main available neural recording circuit and system architectures. Introduction is well written and structured, in general. It would be nice to see few lines about the motivation of providing review of multi-channel neural implants in the presence of previous works. why is it required and what significance will it have on future researchers. Some things provided are too general, for example, the explanation of analog to digital conversion. This can be read in any good text book. It would be better to provide explanation of analog to digital conversion in the context of neural recording implants. There are only 4 papers from 2017 onward. Authors are suggested to search again and include more recent papers. Other than these minor issues, paper seems fine. Best of luck!

Reviewer 3 Report

I think the paper might be a good reference in the literature for those who would like to get a quick overview on Neural implants and bio sensors.

However I suggest to remove or modify the statement on line 36 where it states "Invasive technique are more desirable by most BMIs." To me this is wrong. Signals with non-invasive techniques such as EEG would be more favorable choice for the end users  however they have their own limitation for the obvious reasons mentioned in the manuscript.
